# DECORRELATION SPEEDS UP VISION TRANSFORMERS

## ABSTRACT

Masked Autoencoder (MAE) pre-training of vision transformers (ViTs) yields strong performance in low-label regimes but comes with substantial computational costs, making it impractical in time- and resource-constrained industrial settings. We address this by integrating Decorrelated Backpropagation (DBP) into MAE pre-training, an optimization method that iteratively reduces input correlations at each layer to accelerate convergence. Applied selectively to the encoder, DBP achieves faster pre-training without loss of stability. On ImageNet-1K pre-training with ADE20K fine-tuning, DBP-MAE reduces wall-clock time to baseline performance by 21.1%, lowers carbon emissions by 21.4%, and improves segmentation mIoU by 1.1 points. We observe similar gains when pre-training and fine-tuning on proprietary industrial data, confirming the method's applicability in real-world scenarios. These results demonstrate that DBP can reduce training time and energy use while improving downstream performance for large-scale ViT pre-training.

## 1 INTRODUCTION

Self-attention-based architectures, specifically transformers (Vaswani et al., 2017), have become increasingly popular models both in natural language processing (NLP) and in computer vision. While at first only dominating the NLP domain, transformers have now also become a strong contender in the computer vision domain, competing with or outperforming the previously superior convolutional architectures in tasks such as image classification, segmentation, and detection (Carion et al., 2020; Dosovitskiy et al., 2020).

The transformer's self-attention mechanism lets each patch attend to every other patch in an image, resulting in a true global receptive field. In contrast, convolutional neural networks (CNNs) need to stack many local kernels to gradually grow their receptive field over multiple layers (Parmar et al., 2018; Cordonnier et al., 2019; Ramachandran et al., 2019). However, while convolutional layers scale only linearly with image size, self-attention scales quadratically in the number of image patches, incurring steep compute and memory costs at high resolutions (Vaswani et al., 2017; Dosovitskiy et al., 2020). This overhead, while tolerable in research experiments, becomes a critical bottleneck in industrial pipelines, where GPU hours are limited and rapid processing of imagery (e.g., for defect segmentation) is essential.

Vision transformers (ViTs) are being widely adopted in real-world inspection and diagnostic pipelines, where annotated data is often scarce but high accuracy is critical. Hütten et al. (2022) systematically compared the application of ViT and CNN backbones on railway–freight-car damage assessment imagery and demonstrated that ViTs match or exceed CNN performance in industrial applications with sparsely available data.

Unlike classification, where an image has one label denoting a specific class, segmentation demands pixel-level masks, that classify each pixel as belonging to a particular object. These masks are often difficult to produce, requiring clear boundaries between different objects in the image, and domain-specific experts to perform the annotating. In both industrial-quality control (Božič et al., 2021) and clinical radiology (Bhalgat et al., 2018), this is tiresome and costly work and results in only a few hundred annotated examples per needed object class, even though high-resolution imagery is abundant. This stark contrast between abundant unlabelled imagery and scarce segmentation masks indicates a strong need for representation-learning methods that can leverage large amounts of unlabelled imagery and thus reduce the need for extensive pixel-level annotations.

One promising avenue to leverage abundant unlabelled imagery is Masked Autoencoder (MAE) pre-training (He et al., 2022), which randomly masks a large portion of image patches and trains a ViT to reconstruct the missing pixels. This allows the model to learn rich, generalizable features that transfer well to various downstream tasks when fine-tuned on the limited labelled imagery. However, MAE pre-training on large ViT backbones, such as ViT-Base on ImageNet-1K, can demand tens of GPU-days and hundreds of gigabytes of memory per run, making pre-training on unlabelled data intensive and time consuming (Dosovitskiy et al., 2020; He et al., 2022). On top of the earlier mentioned need for rapid processing of imagery in industrial pipelines, the training of large deep neural networks (DNNs) has also been associated with significant energy consumption and carbon emissions (García-Martín et al., 2019; Strubell et al., 2020; Thompson et al., 2021; De Vries, 2023; Luccioni et al., 2023; 2024). Together, these steep GPU requirements, large memory footprints, and non-trivial carbon emissions highlight why MAE pre-training, while beneficial for large amounts of unlabelled imagery, remains impractical for time- and resource-constrained industrial settings.

The high computational cost of training large neural networks has also motivated research into more efficient optimization methods. Desjardins et al. (2015) showed that whitening layer activations can approximate natural gradient descent, which improves the conditioning of the optimization problem and accelerates convergence. More recently, Ahmad (2024) argued that correlations in layer inputs distort the gradient direction, and that removing these correlations brings standard gradient descent closer to the natural gradient (Amari, 1998). Together, these findings highlight decorrelation as a promising strategy for reducing training cost, making it a natural candidate for addressing the steep computational demands of MAE pre-training.

In this paper we address the impractical training costs of MAE-pretrained ViTs in low-label regimes by applying iterative Decorrelated Backpropagation (DBP) at scale to transformer architectures. DBP, first introduced by Ahmad et al. (2022) and recently adapted to DNNs by Dalm et al. (2024), removes correlations in layer inputs to accelerate convergence. We extend this approach to operate selectively on key blocks of the ViT during MAE pre-training, marking the first large-scale application of DBP in transformer models. This yields: (1) faster pre-training: up to 21.1% wall-time reduction on ImageNet-1K (Deng et al., 2009), (2) improved transfer: higher segmentation mIoU when fine-tuning on ADE20K (Zhou et al., 2017), and (3) real-world impact: similar speed and accuracy gains on proprietary industrial datasets.

We validate our approach with MAE pre-training on ImageNet-1K and downstream segmentation on ADE20K, using randomly sampled 10% subsets to mimic data scarcity. Finally, we demonstrate that DBP-accelerated MAE pre-training offers tangible benefits in industrial vision tasks, providing faster iteration and better fine-tuning performance, while also reducing energy use and carbon emissions.

## 2 METHODS

### 2.1 DECORRELATED BACKPROPAGATION

We employ an iterative decorrelation rule first described in Ahmad et al. (2022). Since decorrelation is applied individually on each layer of a DNN, we first concentrate on describing the decorrelation procedure for one such layer. Given a layer input $\mathbf{x}$ and a decorrelation matrix $\mathbf{R}$, we get the decorrelated input $\mathbf{z}$ as a result of a linear transformation:

$$\mathbf{z} = \mathbf{R}\mathbf{x} \tag{1}$$

This decorrelated input is then used as input for the layer's non-linear transformation

$$\mathbf{y} = f(\mathbf{W}\mathbf{z}) \tag{2}$$

which may be either the kernel function of a convolutional layer applied to a (flattened) input patch or the transformation in a fully-connected layer. Since the application of the decorrelation matrix $\mathbf{R}$ on the correlated input $\mathbf{x}$ is a linear transformation, the full layer transformation can also be described as

$$\mathbf{y} = f(\mathbf{W}\mathbf{R}\mathbf{x}) = f(\mathbf{W}\mathbf{z}) = f(\mathbf{A}\mathbf{x}) \tag{3}$$

with $\mathbf{A} = \mathbf{W}\mathbf{R}$. The output of the layer then becomes the correlated input to the next layer. For each $l$-th layer, $\mathbf{R}_l$ is initially initialised as the identity matrix and gradually learns to decorrelate the

layer inputs during training, using its own, separate, learning rate and optimization scheme, usually vanilla stochastic gradient descent (SGD). This is done in parallel to minimizing the backpropagation (BP) loss during training. In its simplest implementation, $\mathbf{R}_l$ is a square matrix, preserving the dimensionality of the input. After each mini-batch, $\mathbf{R}_l$ is updated as follows:

$$\mathbf{R}_l \leftarrow \mathbf{R}_l - \eta \mathbf{C}_l \mathbf{R}_l \tag{4}$$

where

$$\mathbf{C} = \mathbf{x}_l \mathbf{x}_l^T - \mathrm{diag}(\mathbf{x}_l \mathbf{x}_l^T) \tag{5}$$

and $\eta$ is a small, constant, and tunable learning rate. $\mathbf{x}_l \mathbf{x}_l^T$ is often referred to as the correlation matrix, though technically it is an unnormalized covariance matrix. The objective here is to minimize the off-diagonal elements of $\hat{\mathbf{x}}_l \hat{\mathbf{x}}_l^T$, which represent the covariance structure in the data, without encouraging the trivial solution of setting all activities to be zero. For a derivation of this learning rule, we refer to Ahmad et al. (2022).

## 2.2 APPLICATION TO SCALED VISION TRANSFORMERS

Our MAE pre-training implementation closely mirrors the ViT-Base recipe of He et al. (2022). We divide each image into $16 \times 16$ patches (as in Dosovitskiy et al. (2020)), embed them into a 768-dimensional space, and process them through a 12-layer transformer encoder with 12 attention heads per layer. 75% of these patches are randomly masked during training, and a lightweight, 2-layer decoder reconstructs the missing pixels via an $\ell_2$ loss. He et al. (2022) report no significant difference in fine-tune performance with decoder depths of 2, 4, or 8 layers. We therefore use a 2-layer decoder for maximal efficiency.

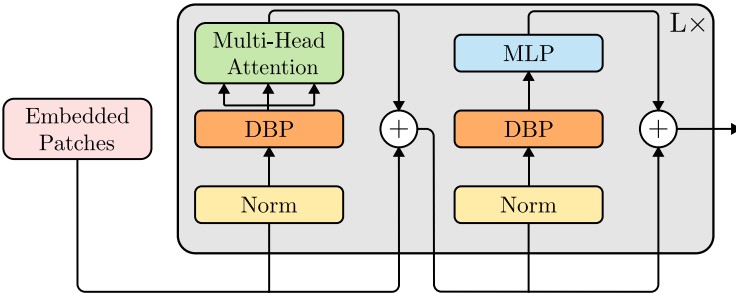

Figure 1: Model overview of DBP application to the vision transformer Encoder. Input images are decorrelated and split into patches, which are then linearly embedded and fed to the encoder. In each of the $L$ encoder blocks, the decorrelation matrix $\mathbf{R}$ is applied to the inputs of the multi-head self-attention layers and to the inputs of the feed-forward MLP layers.

As mentioned in Section 2.1, DBP is applied individually to the layers of a DNN. For the MAE pre-training we apply DBP in the same way, on individual layers in the MAE encoder. Specifically, DBP is applied to the patch-embedding projection, as well as to all the multi-head self-attention layers and the feed-forward MLP layers of each of the 12 encoder layers. Figure 1 depicts an overview of the DBP implementation in the encoder. The decorrelation matrix at each layer is updated at each backward pass according to equation 4, introducing modest computational overhead. On top of this, our early experiments confirmed that applying DBP to the decoder layers destabilized the MAE pre-training (see Appendix B). To both preserve training stability and minimize our computational costs, we apply DBP exclusively to the encoder. This design aligns with our goal to boost encoder representation quality, which is critical for downstream segmentation fine-tuning.

Aside from keeping DBP's computational overhead limited to just the MAE encoder, we also ensure it is limited to just pre-training and does not affect any downstream tasks performed afterwards. Concretely, for each decorrelated layer we compute the fused weight $\tilde{\mathbf{W}} = \mathbf{W}\mathbf{R}$ (see equation 3) and store $\tilde{\mathbf{W}}$ in place of $(\mathbf{W}, \mathbf{R})$. This fusion, performed either at save time or load time, ensures that downstream fine-tuning sees the trained, decorrelated weights directly, without any DBP overhead.

## 2.3 Experimental setup

We compare two pre-training regimes, MAE with DBP ("DBP-MAE") versus MAE with standard backpropagation ("BP-MAE"), to assess both pre-training efficiency and downstream segmentation performance. Each model is pre-trained on a 10% random subset of ImageNet-1K (128,116 images), and then fine-tuned on a 10% random subset of ADE20K (1,021 images). We use 10% subsets to mimic low-label regimes typical of industrial contexts. During MAE pre-training we apply standard data augmentations on the ImageNet-1K images. Each image is randomly resized and center-cropped to our input resolution ($224 \times 224$), with scale sampled uniformly from $[0.2, 1.0]$ and bicubic interpolation. We then apply a horizontal flip with probability 0.5, and normalize pixel values to zero-mean/unit-variance using ImageNet's mean and standard deviation. For fine-tuning on ADE20K, we randomly resize and crop the images to $512 \times 512$ pixels, with its long edge scaled uniformly in $[0.5 \times 512, 2.0 \times 512]$ using bicubic interpolation. We perform a horizontal flip with probability 0.5, followed by random brightness and contrast adjustments ($\pm 25\%$ each) and random hue, saturation, and value shifts ($\pm^{\circ}15$ hue, $\pm 25\%$ saturation/value), mirroring common segmentation augmentations. Finally, we again normalize pixel values to zero-mean/unit-variance statistics using the mean and standard deviation.

For MAE pre-training we adopt similar parameters used by He et al. (2022), training for 1000 epochs with a batch size of 4096. We use the AdamW optimizer (Loshchilov & Hutter, 2017) with parameters $\beta_1 = 0.9$, $\beta_2 = 0.95$, and a weight decay of 0.05. We optimize for the learning rate, finding an optimal learning rate of $5.0 \times 10^{-4}$ for BP-MAE and $1.0 \times 10^{-3}$ for DBP-MAE (see Appendix A). We use a cosine decay learning rate scheduler (Loshchilov & Hutter, 2016) with a warmup of 40 epochs. The pre-trained models are saved at their point of peak performance, measured in validation loss.

During DBP-MAE pre-training we configure three DBP-specific settings to balance stability, performance, and compute. First, we specify which part of the model to decorrelate, which, as explained in Section 2.2, we set to only the encoder of the model. Next, we modify the number of samples used to measure input correlation during the DBP update. Dalm et al. (2024) found that a subsample of 10% is enough for DBP to achieve the same boost in performance, whilst significantly reducing computational overhead. We empirically found this to be true for our experiments as well, so here too we reduce the number of samples for the DBP update to 10% (see Appendix C). Finally, we optimize for the DBP learning rate (see equation 4), finding an optimal learning rate of $5.0 \times 10^{-4}$ for the DBP update (see Appendix A).

Our architecture setup for fine-tuning uses the pre-trained MAE encoder as the ViT-Base encoder, using UperNet (Xiao et al., 2018) as the decoder. We fine-tune for 125 epochs with a batch size of 16, and use the AdamW optimizer with parameters $\beta_1 = 0.9$, $\beta_2 = 0.999$, and a weight decay of 0.05. We optimize for the learning rate, finding an optimal learning rate of $2.5 \times 10^{-5}$, warmed up linearly over the first 500 iterations, then decayed polynomially to zero. Note that we do not apply DBP during fine-tuning. We only differentiate our downstream tasks by their precursor pre-training methods, DBP-MAE or BP-MAE.

We report the pre-training validation loss on the full ImageNet-1K validation set. We validate our fine-tuning performance by reporting the validation loss and the mean Intersection-over-Union (mIoU) on the full ADE20K validation set. Reported wall-clock time was measured as the runtime of the training process during pre-training, excluding validation. Reported carbon emissions was measured as the emissions of the training process during pre-training, excluding validation. All models and algorithms were implemented in PyTorch (Paszke et al., 2019) and run on a compute cluster using Nvidia A100 or H100 GPUs and AMD EPYC 9334 or Intel Xeon Platinum 8360Y processors. Finally, to complement our benchmark results, we evaluate the method, using the same pre-training setup as described above, on proprietary semiconductor wire bonding imagery, thereby assessing its applicability in real-world industrial use cases. This pre-training dataset is comprised of 7,267 samples, of which 7,220 are used for training and 47 are used for validation. Each sample consists of an image of $2 \times 2048 \times 2048$ pixels. For the semantic segmentation downstream task, a similar (but labelled) dataset was used, comprised of 5,560 images for training and 788 images for validation. Again, these sets contain images of $2 \times 2048 \times 2048$ pixels and are annotated for wire, ball, wedge, and epoxy classes. These datasets are provided by a semiconductor company and are not publicly available due to confidentiality agreements.

## 3 RESULTS

### 3.1 DBP SPEEDS UP PRE-TRAINING CONVERGENCE

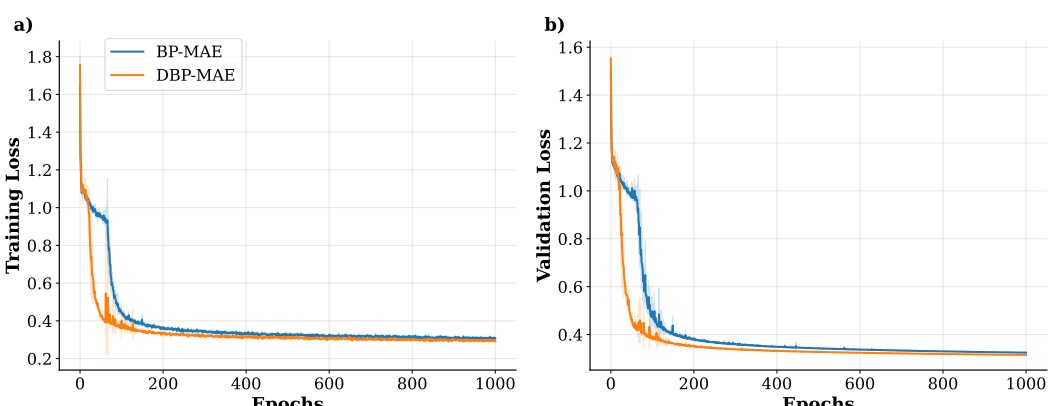

Figure 2: Pre-training performance of BP-MAE and DBP-MAE on ImageNet-1K. Reported results are the average of five random initialized networks where the variability is indicated by the error shading. Panels show training loss (a) and validation loss (b) as a function of the number of epochs. The vertical dotted line indicates the point where DBP is turned off for DBP-MAE.

We observe a clear visual improvement when pre-training MAE on ImageNet-1K with DBP compared to regular BP. Figure 2 shows the training curves for both regimes, where both training loss (Fig. 2a) and validation loss (Fig. 2b) improve faster for DBP-MAE than for BP-MAE, indicating more effective training per epoch. This is most notable when looking at both regimes' initial large drop in both training and validation loss. We see here that DBP-MAE's losses drop in roughly half the number of epochs that it takes for BP-MAE. On top of this visibly faster convergence, both training and validation loss remain consistently better for DBP-MAE than for BP-MAE throughout the full training range. At the end of training, DBP-MAE shows a 2.96% improvement over BP-MAE in terms of validation loss, which is statistically significant with a p-value of $1 \times 10^{-6}$.

An important thing to remember is that DBP comes with some computational overhead per epoch due to the decorrelation update rule (see Equation 4). Considering this, faster convergence as a function of the number of epochs may not mean that DBP-MAE has more efficient training than BP-MAE. For this reason, we also compare pre-training performance for both regimes as a function of wall-clock time in Figure 3[1].

Even though DBP takes 37.3% more time per epoch, it still improves training efficiency over BP in terms of wall-clock time, since we see that the loss drops and converges more quickly and it still maintains a lower training and validation loss throughout the entire runtime. When we take a closer look at the first 10 hours of training in Figures 3c and 3d we can more clearly see the difference in hours for the first major drop in performance, both in training and validation loss. DBP's initial large decrease in loss is done in almost half the time as BP, maintaining the efficiency gain despite the additional compute. Table 1 shows performance levels at different training stages, with DBP achieving 1.3% better validation loss at equal training time, a statistically significant difference with a p-value of $5.8 \times 10^{-4}$.

For concrete efficiency gains of DBP-MAE over BP-MAE we measure the runtime needed for both regimes to reach BP-MAE's peak performance: the lowest achieved validation loss. DBP-MAE is able to reach this performance in 4.6 hours less than BP, giving it a wall-clock time reduction of 21.1% for the same performance (see Figs. 10 and 11a in Appendix D). In terms of energy usage, DBP-MAE is also more efficient than BP-MAE. Measuring the carbon emissions for both regimes to reach BP-MAE's peak performance, DBP-MAE achieves a reduction of 1.49kg or 21.4% (see Fig. 11b in Appendix D).

---

[1]The wall-clock time measurements in Figure 3 were recorded using 2 NVIDIA H100 GPUs and 2 AMD EPYC 9334 CPUs with a (combined) total core count of 64.

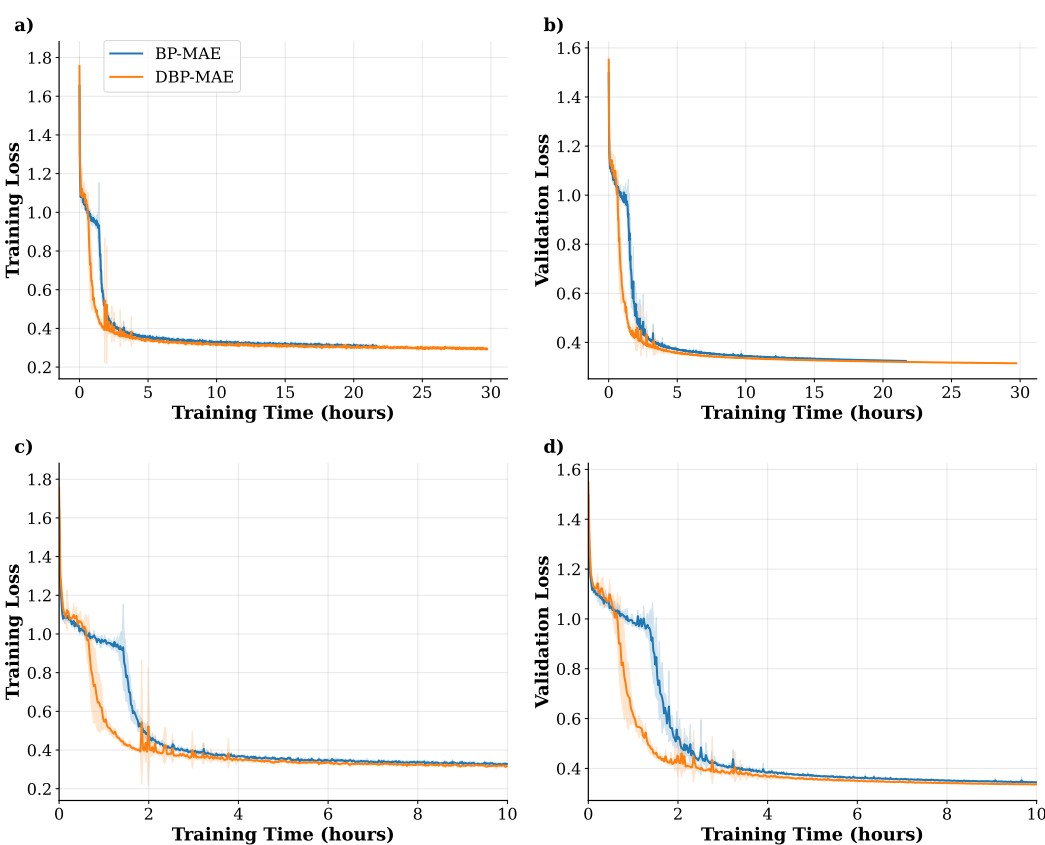

Figure 3: Pre-training performance of BP-MAE and DBP-MAE on ImageNet-1K. Reported results are the average of five random initialized networks where the variability is indicated by the error shading. Panels (a) and (b) show training loss and validation loss respectively as a function of wall-clock time for the full training duration. Panels (c) and (d) show the training loss and validation loss respectively as a function of wall-clock time for the first 10 hours of training.

Table 1: MAE pre-training performance comparison for BP and DBP in terms of training and validation loss. Loss values and corresponding epochs are shown for the regimes' best performance, their final performance, and performance after the same amount of training time. Best results are shown in boldface, with the focus on validation performance.

| Method | Best | | Final | | Equal Time* | |
|---|---|---|---|---|---|---|
| | **Loss** | **Epoch** | **Loss** | **Epoch** | **Loss** | **Epoch** |
| *Validation* | | | | | | |
| BP-MAE | 0.324 | 1000 | 0.324 | 1000 | 0.324 | 1000 |
| DBP-MAE | **0.315** | 994 | **0.315** | 1000 | **0.320** | 728 |
| *Training* | | | | | | |
| BP-MAE | 0.303 | 942 | 0.307 | 1000 | 0.307 | 1000 |
| DBP-MAE | 0.292 | 887 | 0.292 | 1000 | 0.301 | 728 |

*Equal Time Performance: Results when both methods train for 21.7 hours.
DBP improvement at equal time: **+1.3%** validation loss, **+1.9%** training loss.

## 3.2 DBP PRE-TRAINING INCREASES FINE-TUNE PERFORMANCE

Figure 4 shows the downstream task performance using pre-trained DBP-MAE and BP-MAE. Here too, we observe clear visual differences when comparing the performance between the two regimes

of semantic segmentation fine-tuning on ADE20K. While there are no differences in terms of efficiency or speed, since no DBP is applied during fine-tuning, we do see differences in the achieved performance values. First, looking at the training loss (Fig. 4a) we see that fine-tuning on DBP-MAE results in a slightly lower achieved loss, indicating better training convergence.

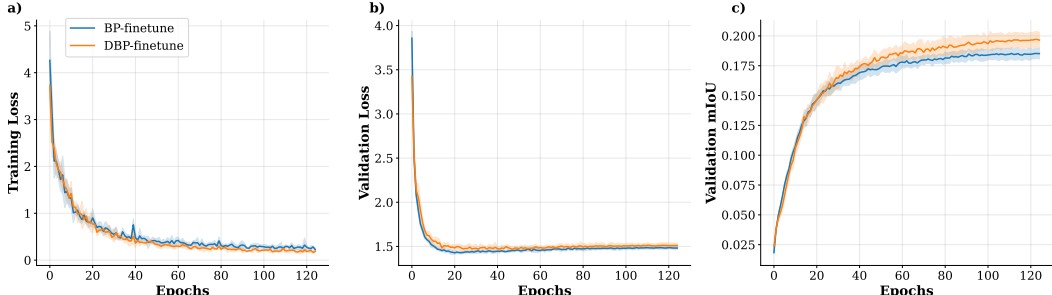

Figure 4: Performance comparison of semantic segmentation fine-tuning on ADE20K, using pre-trained DBP-MAE and BP-MAE. Reported results are the average of fine-tuning on five random initialized pre-trained network checkpoints where the variability is indicated by the error shading. Panels show performance for the two regimes in training loss (a), validation loss (b), and validation mean IoU (c) as a function of the number of epochs.

Second, when we look at validation loss (Fig. 4b) we see that fine-tuning on BP-MAE actually performs better than on DBP-MAE, resulting in a slightly lower loss, indicating better generalization. However, this difference is not statistically significant, with a p-value of $0.069$. Finally, when we look at the validation mIoU (Fig. 4c), we observe a performance increase when fine-tuning on DBP-MAE compared to fine-tuning on BP-MAE, with the mIoU curve already increasing earlier during fine-tuning and ending with a significantly higher performance, a $6.11\%$ improvement with a p-value of $0.013$. Overall, the validation mIoU performance increase indicates that pre-training with DBP offers a downstream task performance boost and leads to better semantic segmentation results, achieving a mIoU of $0.196$ compared to BP-MAE's fine-tuning mIoU of $0.185$.

### 3.3 DBP PROVIDES SPEED-UP AND PERFORMANCE GAINS ON INDUSTRY DATA

When comparing the results of the same experiment on the semiconductor wire bonding data, we observe a similar performance gain when using DBP during pre-training, compared to regular BP. Figure 5 shows that here too, both training and validation loss drop much quicker for DBP-MAE, and remain consistently lower than BP-MAE throughout pre-training. DBP achieves a significant $39.8\%$ validation loss improvement at the end of training ($p = 0.001$) and a significant $46.4\%$ validation loss improvement at its peak ($p = 0.0006$).

Figure 6 shows the downstream task performance, comparing results of fine-tuning on BP-MAE and DBP-MAE. We observe a jittery training loss (Fig. 6a) where fine-tuning on DBP results in a slightly lower loss throughout training than fine-tuning on BP. The DBP validation loss (Fig. 6b) is also better throughout training, again showing a slightly lower loss throughout the process and achieving a significant improvement of $24.12\%$ over BP ($p = 0.0001$). Lastly, we again observe a higher DBP validation mIoU (Fig. 6c) compared to BP, resulting in a significant improvement of $16.38\%$ ($p = 0.0002$). This indicates that again, applying DBP during pre-training results in a better downstream task performance.

Overall, we observe similar benefits of applying DBP during pre-training on industrial data as seen on the benchmark datasets, indicating that DBP maintains the same gains in applicable real-world industrial fields like the semiconductor industry.

## 4 DISCUSSION

Real-world industrial inspection pipelines often suffer from the lack of available, high-quality annotated segmentation data to properly train DNNs via supervised learning. A solution to this is to use

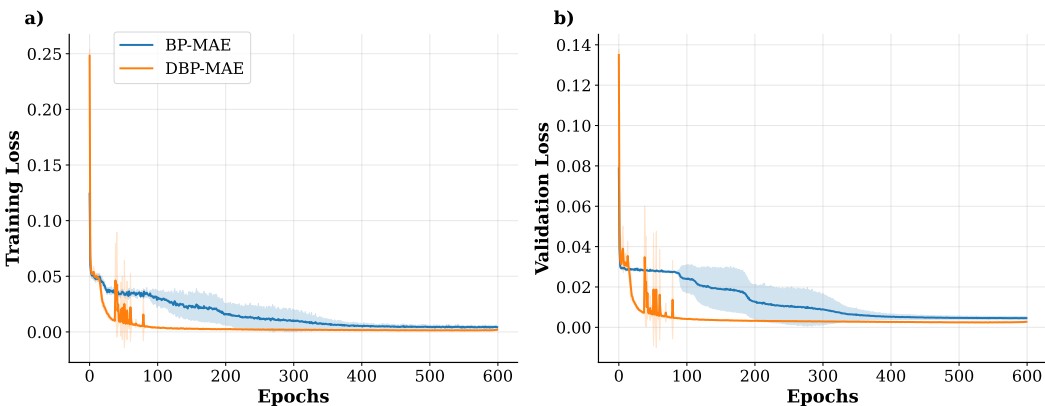

Figure 5: Pre-training performance of BP-MAE and DBP-MAE on semiconductor wire bonding data. Reported results are the average of five random initialized networks where the variability is indicated by the error shading. Panels show training loss (a) and validation loss (b) as a function of the number of epochs.

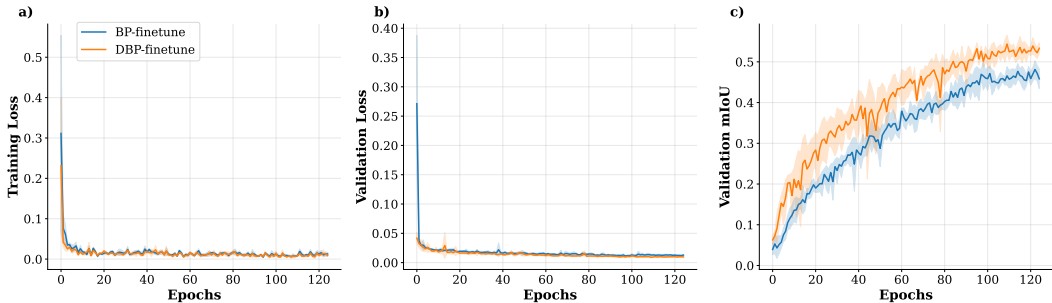

Figure 6: Performance comparison of semantic segmentation fine-tuning on semiconductor wire bonding data, using pre-trained DBP-MAE and BP-MAE. Reported results are the average of fine-tuning on five random initialized pre-trained network checkpoints where the variability is indicated by the error shading. Panels show performance for the two regimes in training loss (a), validation loss (b), and validation mean IoU (c) as a function of the number of epochs.

self-supervised learning instead, using methods like MAE pre-training without labelled data, and then perform downstream tasks using the (little) annotated data that is available. However, training on large ViT backbones demands large amounts of compute and time, as well as introducing significant energy consumption and carbon emissions. In this paper, we have shown that decorrelated backpropagation provides a suitable solution for more efficient deep learning. Results show that, by applying DBP to MAE pre-training on ImageNet-1K, we can achieve the same performance as regular BP but with a runtime reduction of 21.1%. Additionally, we achieve better pre-training performance when training for the same amount of time or longer, achieving a significantly lower validation loss. We also demonstrate our method to be more energy efficient, showing a reduction in carbon emissions of 21.4% when training to the same performance as BP. Our results show that, despite the increased computational load per epoch, DBP helps the model train and converge faster, as well as reducing energy consumption and carbon emissions.

We also demonstrated how applying DBP during pre-training leads to better downstream task performance. When fine-tuning on ADE20K, DBP pre-training resulted in a higher validation mIoU, indicating better semantic segmentation, a crucial metric for real-world applications. While both training loss and validation mIoU performed better using DBP pre-trained models, the fine-tune validation loss performance was less than that of regular BP pre-trained models. We suspect this is caused by the inherent difference between how the validation loss (cross-entropy) and mIoU are calculated. Whereas cross-entropy is probabilistic and pixel-wise, mIoU is a better indication of learned visual representations and segmentation performance. Especially for practical applications

we are more interested in the latter, as it directly indicates how well the model performs on the real-world applications that they are used for.

Finally, we expanded our benchmark results by showing that DBP maintains the same performance gains in real-world, industrial applications. We show that when pre-training on a semiconductor wire bonding image dataset, DBP shows the same increase in training and convergence speed as before, as well as achieving better peak performance. When fine-tuning for semantic segmentation on the semiconductor data, here too we see that the DBP pre-trained model is able to achieve higher downstream task performance. This indicates that DBP allows industrial inspection pipelines to train faster, and on top of that can boost segmentation performance, which is of crucial to industries where demand for efficiency and dependable performance is a priority.

In addition to the semiconductor industrial setting, we believe that the DBP-MAE self-supervised technique could also be highly beneficial for medical imaging segmentation. While the medical domain differs substantially from the semiconductor industrial field and presents unique challenges, it similarly suffers from limited datasets and costly, expert-driven annotations. This could be useful particularly for 3D radiological CT and MR scans or large 2D digital pathology slides, where faster pre-training and stronger segmentation performance are critical. Since variants of MAE pre-training have already been applied successfully to medical datasets (Zhou et al., 2023; Liu et al., 2024; Tang et al., 2025), accelerating them and further enhancing performance with DBP-MAE represents a promising direction for future research.

Alongside our results, we note several limitations and avenues for future work. First, we evaluate DBP only on 10% random subsets of the data (ImageNet-1K and ADE20K). While this mimics data scarcity in industrial and medical domains, it precludes direct comparison of DBP performance to full-data benchmarks. We compare against BP-MAE with the same 10% subset settings, providing a valid baseline for the effects of DBP, but we cannot yet generalize to larger-scale pre-training.

Second, the storing and updating of decorrelation matrices required by DBP introduces some additional compute and memory requirements. This is represented by the increased wall-clock time per epoch during training. To reduce this additional computational overhead even further, more effective use could be made of the sparseness structure of the decorrelation matrix. For instance, the large decorrelation matrix could be approximated and updated with the use of low-rank matrix factorization (Eckart & Young, 1936; Srebro & Jaakkola, 2003).

Third, we observed that pre-training MAE on ImageNet-1K can be done for longer, but doing so results in diminishing returns as the loss only decreases by a fraction more, for an additional training duration of several hours. When continuing pre-training DBP-MAE, we also observed large instabilities in training, with both training and validation loss increasing near the end. This is most likely caused by the regular learning rate scheduler becoming too small, compared to the DBP learning rate which does not have a scheduler. If one would want to pre-train MAE for longer and retain stable learning throughout the process, we suggest either implementing a similar scheduler for the DBP learning rate or an early stopping mechanism for the DBP update.

Finally, the selection of the model architecture to apply DBP on, in our case the encoder, was found empirically. We observed pre-training performance and convergence speed increase when applying DBP to the encoder as compared to our BP baseline. When applying DBP to the decoder (either also or exclusively), we observed pre-training becoming unstable and even failing to converge (see Appendix B). Future work could develop automated selection methods for DBP, that could automatically determine which parts of the model, or even which specific layers, would benefit most from DBP. Such methods could further reduce computational overhead and result in faster training, reduced energy use, and lower carbon emissions.

In this work, we demonstrated that we can increase training efficiency and boost performance for large vision transformers, both on benchmark datasets and on real-world industrial data. We expect that additional work on the application of DBP to such networks will yield even larger reductions in convergence speed, and hope to inspire large industries to strive for further reductions in energy consumption and carbon emissions.

## CODE AVAILABILITY

The code for training and evaluating the decorrelated backpropagation algorithm is available here.

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

## A  HYPERPARAMETER OPTIMIZATION

We performed a grid search over the learning rates for **R** and **W** to find the parameter values that yield optimal performance for each algorithm, BP-MAE and DBP-MAE. Both the training loss and validation loss were compared after 70 epochs of MAE pre-training, to get an early indication of performance without wasting too many resources on optimization. Based on the grid search

outcomes, initially a $\mathbf{W}$ learning rate of $3.0 \times 10^{-4}$ was chosen for BP and $1.0 \times 10^{-3}$ for DBP, together with a $\mathbf{R}$ learning rate of $2.5 \times 10^{-4}$, as these yielded the lowest training loss values after 70 epochs. However, observing the training curves and later performance revealed that for BP, a $\mathbf{W}$ learning rate of $5.0 \times 10^{-4}$ yielded slightly better performance and more stable training. Observing the validation loss after 70 epochs for DBP reveals that a $\mathbf{R}$ learning rate of $5.0 \times 10^{-4}$ gives a better generalization performance, so we chose that for the $\mathbf{R}$ learning rate instead. Figure 7 shows the grid search outcomes.

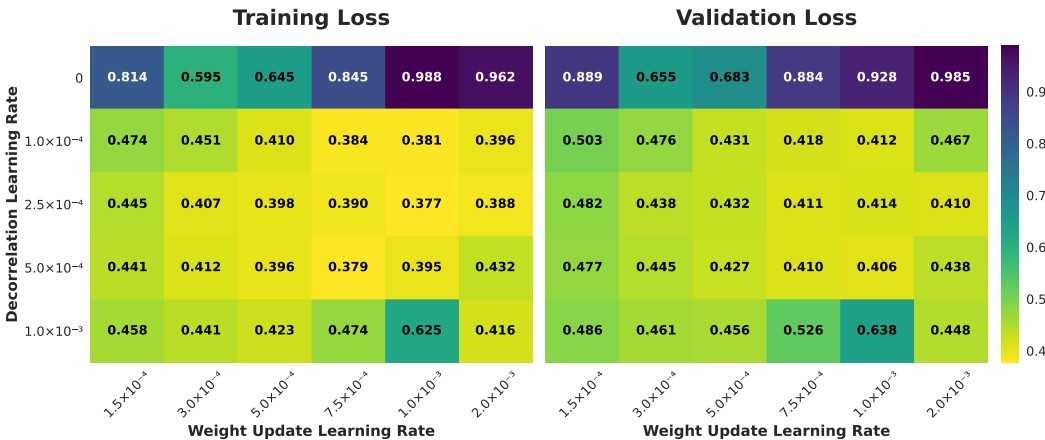

Figure 7: MAE pre-training performance after 70 epochs, measured in training loss (left) and validation loss (right) for various weight learning rates and decorrelation learning rates. Regular BP performance is represented as a decorrelation learning rate of zero.

## B  MODULE SELECTION FOR DBP

For the selection of which modules to apply DBP to during MAE pre-training, we performed a comparison experiment of 200 epochs, using the same optimal parameters as for our main results, to find the performance when decorrelating only the encoder, compared to decorrelating the full model (both encoder and decoder). We found that when additionally decorrelating the decoder, training becomes very unstable and performance measured in both training loss and validation loss is worse, compared to only decorrelating the encoder. This result is not entirely unexpected, as inherently a DNN's decoder has lower to no input correlations, as it does not process actual data like an encoder does. We expect that this is why decorrelating the decoder does not yield any performance improvements, as there is little correlations in the inputs to reduce. On top of this, decorrelating might actually harm learning and result in the observed instability, as DBP might be affecting the weights too much for stable learning. Figure 8 shows the comparison in performance between the module selections.

## C  DBP SUBSAMPLE SIZE

To find the optimal sample size used to measure input correlation during the DBP update, we performed a comparison experiment of 200 epochs, using the same optimal parameters as for our main results. Figure 9 shows the performance comparison for sample sizes of $10\%$, $20\%$, and $5\%$. We found that a sample size of $5\%$ can introduce some instability during training, suggesting that using too small of a sample to measure input correlations reduces the effectivity of DBP. We also found that sample sizes of $10\%$ and $20\%$ showed no visible difference in performance, whilst requiring twice the amount of memory to calculate input correlations. Additionally, when attempting to train the model with a higher DBP sample size, namely $50\%$, we encountered out-of-memory errors, suggesting that such large sample sizes are not feasible to use for DBP to remain efficient. Based on these findings, we selected a subsample size of $10\%$ to measure input correlations during the DBP update, maintaining performance and effectivity whilst reducing memory usage as much as possible.

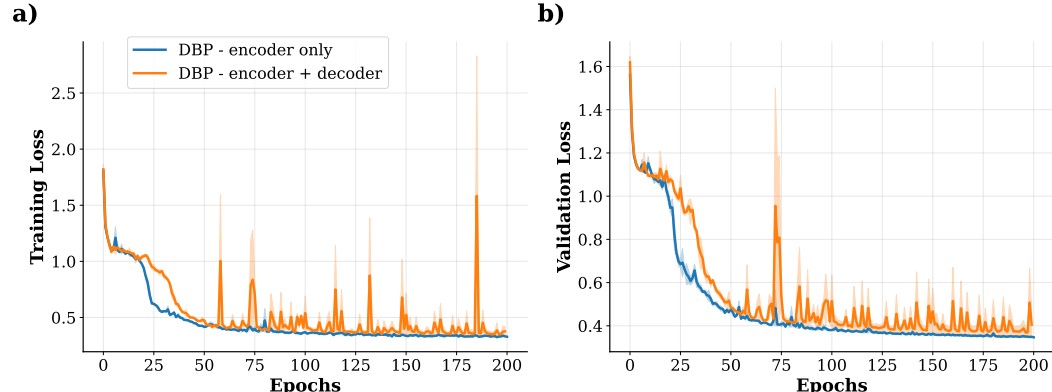

Figure 8: Pre-training performance for DBP-MAE on ImageNet-1K with DBP applied to only the encoder, and to the full model. Reported results are the average of 2 random initialized networks where the variability is indicated by the error shading. Panels show training loss (a) and validation loss (b) as a function of the number of epochs.

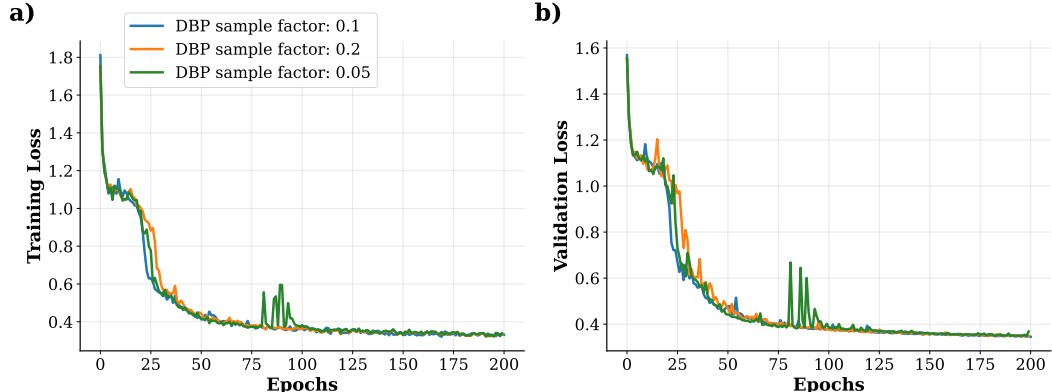

Figure 9: Pre-training performance for DBP-MAE on ImageNet-1K with different sample sizes used to measure input correlation during the DBP update. Panels show training loss (a) and validation loss (b) as a function of the number of epochs.

## D   RUNTIME AND CARBON EMISSION REDUCTION

Figure 10 shows the two points where BP-MAE and DBP-MAE achieve the max performance of BP-MAE. It also indicates the difference in performance in hours, showing that DBP saves 4.6 hours of wall-clock time to achieve BP-MAE top performance.

Figure 11a again shows the difference in wall-clock time needed to achieve BP-MAE top performance for both BP-MAE and DBP-MAE, and highlights the difference between them. Figure 11b shows a similar difference, but in $CO_2$ emissions in kg. Here too it indicates the difference, showing that DBP saves 1.49kg of carbon emissions to achieve BP-MAE top performance.

## E   COMPARISON OF INPUT CORRELATIONS

Figure 12 shows the average input correlations (or decorrelation loss) during pre-training on ImageNet-1K for BP and DBP. It shows that even though BP learns to reduce correlations in the data over time, DBP has an enormously larger reduction quite early on, showing the effectiveness of the decorrelation algorithm.

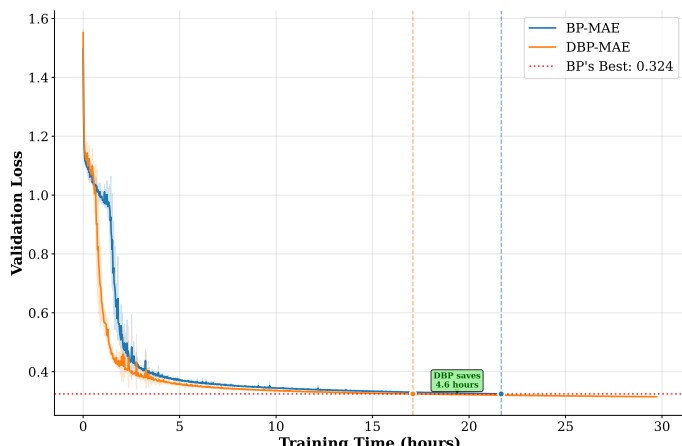

Figure 10: Pre-training performance of BP-MAE and DBP-MAE on ImageNet-1K. Reported results are the average of five random initialized networks where the variability is indicated by the error shading. Figure highlights the points in training time where BP-MAE and DBP-MAE reach BP-MAE's peak performance, and the difference between these points in hours.

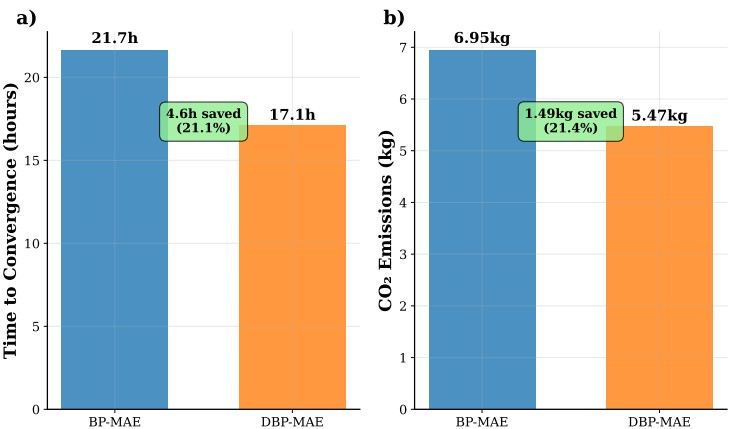

Figure 11: Efficiency gains of DBP-MAE over BP-MAE pre-training on ImageNet-1K. a) shows the time in hours to convergence to BP-MAE's peak performance, for both regimes. b) shows the $CO_2$ emissions in kg to convergence to BP-MAE's peak performance, for both regimes.

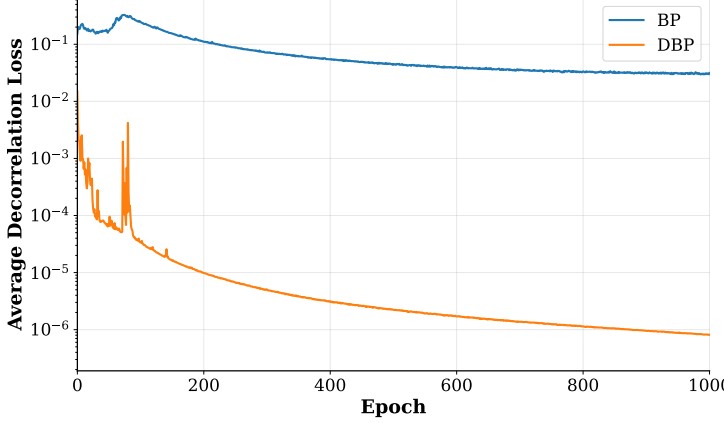

Figure 12: Comparison of layer input correlations during pre-training on ImageNet-1K averaged over the network layers, as a function of the number of epochs (logarithmic scale).

