# OpenReview forum: "Decorrelation Speeds Up Vision Transformers"
_ICLR.cc/2026/Conference — ICLR 2026 Conference Withdrawn Submission_

### Official Review · Reviewer_mchK · 2025-10-23

**Soundness:** 3
**Presentation:** 3
**Contribution:** 2
**Rating:** 4
**Confidence:** 3

**Summary:**

This paper applies DBP to vision transformers for MAE pre-training to reduce computational costs. DBP reduces correlations in layer inputs during training, which allows faster convergence and reduces the total training time needed. Experiments on 10% ImageNet-1K subsets show DBP-MAE achieves reasonable wall-clock time reduction and carbon emission reduction compared to BP-MAE. Downstream tasks including ADE20K segmentation and semiconductor wire bonding data show reasonable performance improvements.

**Strengths:**

- The paper has clear motivation. MAE pre-training is expensive and this work provides a solution to reduce training time and energy consumption for industrial applications.

- The experimental results show reasonable improvements.

**Weaknesses:**

- This paper applies existing DBP to MAE training but offers limited technical novelty. The core DBP method comes from prior work and this paper mainly demonstrates faster convergence on MAE. The technical contribution description is confined to only Section 2.2.

- The convergence analysis is insufficient. This paper claims DBP works better because validation loss drops faster during MAE pre-training. However validation loss is only an indirect measure of learned features. To prove DBP actually learns better representations the paper should test classification performance at fixed training intervals by freezing MAE weights and training linear classifiers on ImageNet.

- The evaluation scope is too limited. This paper uses only 10% ImageNet data to simulate industrial scenarios but this severely restricts generalizability claims. The industrial validation uses only semiconductor wire bonding data which is insufficient to show broad applicability. The 10% subset cannot prove DBP outperforms BP in general settings.

- Equation (5) defines C as $x x^T - \text{diag}(x x^T)$ which is nonstandard and dimensionally ambiguous. The diag operation changes matrix dimensions and the subtraction is ill-defined. Please clearly explain this computation.

**Questions:**

Please refer to weaknesses. Overall this paper has insufficient technical contribution as it mainly applies DBP technology. However the experimental analysis and evaluation are insufficient to support the non-technical contributions.

---

### Official Review · Reviewer_hCLK · 2025-10-30

**Soundness:** 2
**Presentation:** 2
**Contribution:** 2
**Rating:** 2
**Confidence:** 4

**Summary:**

This paper integrates **Decorrelated Backpropagation (DBP)** — a gradient orthogonalization technique previously proposed by Ahmad et al. (2022) and Dalm et al. (2024) — into **Masked Autoencoder (MAE)** pre-training of Vision Transformers (ViT).
By applying layer-wise decorrelation to the ViT encoder during MAE training, the authors claim:

- **21.1% wall-clock reduction** to reach the same validation loss as the baseline MAE,
- **21.4% lower CO₂ emissions**,
- **+1.1 mIoU** improvement when fine-tuned on ADE20K,
- Similar trends on a proprietary industrial dataset.

The method requires no architectural modification and can be fused into pretrained weights post-training.

**Strengths:**

1. **Practical relevance.** Tackles a concrete bottleneck: the high cost and energy footprint of ViT pre-training.
2. **Implementation simplicity.** DBP can be easily inserted into existing MAE pipelines and removed after training.
3. **Comprehensive experiments.** The authors test both on academic benchmarks (ImageNet-1K / ADE20K) and industrial datasets, including energy usage statistics.
4. **Well-structured presentation.** The methodology and appendices are clear and reproducible.
5. **Open-sourced code** ensures verifiability.

**Weaknesses:**

1. **Limited novelty.** The method is a direct application of prior DBP work to MAE; no new algorithmic or theoretical contribution.
2. **Insufficient causal evidence.** Faster convergence might stem from larger learning rates, not decorrelation; gradient-level analysis is missing.
3. **Restricted experiment scale.** Only 10% subsets of ImageNet and ADE20K are used; results may not generalize to full datasets.
4. **Compute metrics incomplete.** “Speed-up” ignores per-epoch overhead; CO₂ reduction is marginal and poorly defined.
5. **Weak downstream gains.** The ADE20K mIoU gain (+1.1) is minor, and validation loss is worse than baseline.
6. **Overclaiming.** The title and abstract exaggerate the impact; results mainly show modest empirical acceleration in small-scale settings.
7. **Scaling to larger models.** No results on ViT-L, and -H models.

**Questions:**

1. How does DBP interact with LayerNorm or adaptive optimizers (AdamW, LAMB)?
2. Could the observed gains be reproduced under identical compute budgets or full datasets?
3. How were carbon emissions measured?
4. Does decorrelation affect representation diversity or anisotropy?

---

### Official Review · Reviewer_iSxr · 2025-10-31

**Soundness:** 2
**Presentation:** 2
**Contribution:** 2
**Rating:** 2
**Confidence:** 4

**Summary:**

This work addresses the significant computational cost associated with Masked Autoencoder (MAE) pre-training for Vision Transformers (ViTs). The authors posit that while MAE is a powerful technique for leveraging large amounts of unlabeled data in industrial settings (e.g., segmentation tasks where labeled data is scarce), its high resource demand in terms of GPU time and energy consumption makes it impractical for many real-world applications.

To address this bottleneck, the authors propose integrating Decorrelated Backpropagation (DBP) into the MAE pre-training process. The core technical claim is that DBP, which is applied selectively to the ViT's encoder, accelerates convergence by iteratively reducing statistical correlations in the layer inputs. The authors claim that the method improves the conditioning of the optimization problem, linking it to prior work on natural gradient descent.

The authors evaluate their method, DBP-MAE, by pre-training on a random subset of ImageNet-1K and fine-tuning on the ADE20K segmentation benchmark to mimic scenarios with low data regime. They report that their method reduces the wall-clock pre-training time to reach baseline performance by 21.1\% while also improving the downstream mIoU by 1.1\%. The authors also note a corresponding 21.4% reduction in carbon emissions and state that these efficiency and accuracy gains were replicated on proprietary industrial datasets.

**Strengths:**

The following are some positives of this work:
- The paper tackles a highly relevant problem of improving training efficiency. The computational cost of pre-training large models like MAE-ViTs is a significant, practical bottleneck for both industry and academia. The focus on industrial applications with scarce labeled data (like defect segmentation) is well-motivated.
- I really enjoyed the face that the  authors not only measure reduction in wall-clock time but also explicitly quantifying the reduction in carbon emissions (21.4%).
- The authors also show improvement over baselines which is interesting given many works on improving transformer efficiency show performance lower or comparable than the baseline.

**Weaknesses:**

The following are the concerns I have with this work:

- My confusion lies in the core motivation. The text describes $z = Rx$ as a decorrelated input (Eq. 1) which is then immediately used as input to the next transformation $y = f(Wz)$ (Eq. 2). As the authors correctly state, this is equivalent to $y = f(WRx) = f(Ax)$ (Eq. 3). The output y, which is the only input to the next layer (e.g., the Multi-Head Attention block in your figure) is the result of the full, re-correlated transformation $A=WR$. The decorrelated representation $z$ is a transient internal variable that is never used outside the DBP block. If the final output y is not decorrelated, can the authors please clarify what is the precise benefit of this method?
- I also do not understand how the DBP block adds an overhead during training, but not during inference. From Figure 1, the DBP block introduced additional trainable parameters $\tilde{W}$ introducing overhead during training. But $\tilde{W}$ is also present during inference. So as the authors claim, why does it not lead to overhead during inference.
- In L167, the authors say that "each model is pretrained on 10% *random* subset of Imagenet-1K". If this is the case, i think that it is this randomness that leads to faster convergence and not the DBP block. The reason being that if BDP-MAE model gets a more easier subset during pretraining, then it is not surprising that it will lead to faster convergence as shown in Figure 2. Also Figure 2 seems quite surprising that training loss and validation loss have the exact same curve and magnitude.  Can the authors please ensure there is no error in the plots.
- Instead of measuring training efficiency in terms of wall-clock time, can the authors please report efficiency in terms of FLOPs, throughput (images/sec). Also instead of comparing to just baseline MAE, it would be beneficial to compare against methods that improve efficiency such as Token Merging [Bolya et al.], [1*, 2*, 3*, 4*]. Interestingly [1*, 2*] also decorrelate the features, so its more similar to the direction of this research work.

[1*] Zhou et al., Refiner: Refining Self-attention for Vision Transformers
[2*] Venkataramanan et al., Skip-Attention: Improving Vision Transformers by Paying Less Attention, ICLR 2024
[3*] Zhang et al., Depth-Wise Convolutions in Vision Transformers for Efficient Training on Small Datasets
[4*] Han et al., On the connection between local attention and dynamic depth-wise convolution, ICLR 2022
[5*] Fayyaz et al., Adaptive token sampling for efficient vision transformers, ECCV 2022
[6*] Tang et al., Patch slimming for efficient vision transformers, CVPR 2022

**Questions:**

- Can the authors please provide qualitative analysis showing correlation in input layers, which DBP block decorrelated. I suggest the authors to look into an interesting analysis in [2*], where the work using CKA to measure correlated across different layers in a vision transformer.

---

### Official Review · Reviewer_8yvx · 2025-10-31

**Soundness:** 2
**Presentation:** 2
**Contribution:** 1
**Rating:** 2
**Confidence:** 4

**Summary:**

In this paper, the authors propose DBP-MAE, a method designed to reduce the computational cost of MAE pretraining through decorrelated backpropagation. The core idea of the DBP block is to decrease the correlation between each block’s input features, making them more independent. This reduces gradient interference and improves training stability. Experiments demonstrate that DBP-MAE achieves comparable performance to the original MAE while reducing training time by 21.1% and carbon emissions by 21.4%.

**Strengths:**

- The idea is simple, intuitive, and easy to follow.
- Experimental results show faster convergence compared to the original MAE, with improvements observed on segmentation tasks.

**Weaknesses:**

- The paper lacks a systematic related work section. The concept of decorrelated backpropagation has been explored extensively in prior works (e.g., [1]), especially for CNNs. Without a deeper theoretical or empirical distinction, the contribution seems incremental, mainly adapting an existing idea to a new backbone.

- The evidence provided is insufficient to convincingly demonstrate that the proposed method accelerates training. Validation loss alone is not always a reliable indicator of training efficiency, as it may not correlate strongly with downstream performance. Furthermore, prior studies (e.g., [2]) have shown that MAE itself can already train efficiently and achieve strong results.

- The experimental scale is too limited for a pretraining method. A full ImageNet pretraining is necessary for a fair evaluation. The current experiments focus only on segmentation tasks, while standard benchmarks such as classification are missing. Achieving similar performance as the original MAE on a larger, standardized setting would make the claim much more convincing.


[1] Dalm, Sander, et al. "Efficient deep learning with decorrelated backpropagation." *arXiv preprint arXiv:2405.02385* (2024).


[2] Wei, Zihao, et al. "Masked autoencoders are secretly efficient learners." *Proceedings of the IEEE/CVF Conference on Computer Vision and Pattern Recognition*. 2024.

**Questions:**

- Can the authors provide clearer comparisons with previous decorrelation-based methods to highlight the novelty of their approach?

- Is it possible to conduct experiments at the same scale as the original MAE to verify scalability and consistency?

---

### Note · Authors · 2025-11-21

I have read and agree with the venue's withdrawal policy on behalf of myself and my co-authors.